# AMPK: Regulation of Metabolic Dynamics in the Context of Autophagy

**DOI:** 10.3390/ijms19123812

**Published:** 2018-11-29

**Authors:** Isaac Tamargo-Gómez, Guillermo Mariño

**Affiliations:** 1Instituto de Investigación Sanitaria del Principado de Asturias, 33011 Oviedo, Spain; isaactamargo13@gmail.com; 2Departamento de Biología Funcional, Universidad de Oviedo, 33011 Oviedo, Spain

**Keywords:** AMPK, autophagy, metabolism, mTOR, ULK

## Abstract

Eukaryotic cells have developed mechanisms that allow them to link growth and proliferation to the availability of energy and biomolecules. AMPK (adenosine monophosphate-activated protein kinase) is one of the most important molecular energy sensors in eukaryotic cells. AMPK activity is able to control a wide variety of metabolic processes connecting cellular metabolism with energy availability. Autophagy is an evolutionarily conserved catabolic pathway whose activity provides energy and basic building blocks for the synthesis of new biomolecules. Given the importance of autophagic degradation for energy production in situations of nutrient scarcity, it seems logical that eukaryotic cells have developed multiple molecular links between AMPK signaling and autophagy regulation. In this review, we will discuss the importance of AMPK activity for diverse aspects of cellular metabolism, and how AMPK modulates autophagic degradation and adapts it to cellular energetic status. We will explain how AMPK-mediated signaling is mechanistically involved in autophagy regulation both through specific phosphorylation of autophagy-relevant proteins or by indirectly impacting in the activity of additional autophagy regulators.

## 1. Introduction

Eukaryotic cells are able to adapt to adverse fluctuations in the cellular environment. In order to do so, cells have developed molecular sensors, which react to circumstances that may perturb cell homeostasis. AMPK (adenosine monophosphate-activated protein kinase) is one of the main cellular sensors able to link a variety of cellular functions and processes to energy availability. AMPK is an evolutionarily conserved protein kinase, present both in unicellular organisms, such as baker’s yeast, and also in more complex multicellular eukaryotes, as mammals. In 1981, the role for AMPK yeast ortholog, SNF1 (Sucrose Non-fermenting 1) as the main energy sensor in this organism was described [1]. This kinase is responsible for activating alternate metabolic pathways when the main carbon or nitrogen sources change, thus adapting cellular metabolism to oscillations in the cellular environment. In higher eukaryotes, AMPK is activated when AMP (adenosine monophosphate):ATP (adenosine triphosphate) and/or ADP (adenosine diphosphate):ATP ratios increase [1,2]. Once activated, AMPK maintains energy homeostasis by two complementary actions, an inhibition of ATP-consuming anabolic processes coupled with an activation of ATP-generating catabolic processes [3]. Due to its importance for cellular energy homeostasis, the activity of AMPK is tightly controlled by multiple upstream regulators, which contributes to link cellular metabolism with oscillating parameters in the cellular environment and also with changes in cellular nutritional and energetic requirements [4]. Moreover, AMPK signaling pathways are involved in numerous physiological processes apart from their main metabolic functions, such as cytoskeleton remodeling and transcriptional control or regulation of essential cellular processes, such as apoptosis or autophagy [5].

Autophagy is the cellular process by which organelles, proteins, and different macromolecules are delivered to the lysosomes for degradation [6]. This process can be classified into at least three different pathways: Macroautophagy (which we will refer to as autophagy), microautophagy, and chaperone-mediated autophagy, which mainly differ in the way in which autophagic cargo is transferred to the lysosome [7,8,9]. Autophagy plays essential roles in all eukaryotic cells and has been implicated in multiple processes, such as cell differentiation, cell death, and regulation of innate and adaptive immune responses or antigen presentation, among many other processes in high eukaryotes [10,11]. Despite all these functions, autophagy’s most evolutionarily conserved role, from yeast to mammals, is to sustain energy balance in the cell by providing ATP and building blocks (lipids, amino acids, nucleotides, etc.) out of the degradation of non-essential or damaged cellular structures. Although autophagy regulation is complex and a variety of signaling cascades and regulatory mechanisms modulate autophagic activity, AMPK is probably the most conserved autophagy inducer through evolution. AMPK activity is linked to autophagic degradation in almost all eukaryotic cells.

In this review, we describe the main metabolic regulatory functions of AMPK, including its prominent role in autophagy regulation. We will discuss how AMPK activity is key to the coordination of catabolic pathways which produce energy and micro-molecules with anabolic processes, which use energy and micro-molecules to synthesize new macro-molecules which may be essential to sustain cell viability when cells face significant alterations in the intracellular/extracellular environment.

## 2. AMPK: Structure and Activation Mechanism

Evolutionarily, AMPK is a highly conserved serine/threonine protein kinase, and a member of the AMPK-related kinase family, which is comprised of thirteen kinases in the human genome. In mammalian cells, AMPK exists as a heterotrimeric complex formed by a catalytic α subunit and regulatory β and γ subunits [12]. There are multiple isoforms for each subunit encoded by different genes, *PRKAA* (5′-AMP-activated protein kinase catalytic subunit alpha), *PRKAB* (5′-AMP-activated protein kinase subunit beta) and *PRKAG* (5′-AMP-activated protein kinase subunit gamma). In humans, there are seven genes encoding AMPK subunits: Two isoforms for the α subunit (α1 and α2), encoded by the genes *PRKAA1* and *PRKAA2*, two isoforms of the β subunit (β1 and β2), encoded by *PRKAB1,* and three isoforms of the γ subunit (γ1, γ2 and γ3), encoded by *PRKAG1*, *PRKAG2* and *PRKAG3,* respectively [13]. Each AMPK complex is composed by one α-subunit, one β-subunit, and one γ-subunit (Figure 1). The fact that all combinations are possible leads to twelve different conformations in which α, β and γ subunits may constitute a functional AMPK complex, which are normally associated with a specific tissue or a determined cell type, or determined subcellular localizations inside cells [14].

The α-subunit presents a serine/threonine kinase domain at the N-terminal region and a critical residue, Thr172. This conserved residue can be phosphorylated by several different upstream kinases, which constitutes the main mechanism by which AMPK activity is regulated at the short term. Different groups have shown that the LKB1 (Liver kinase B1) kinase is able to phosphorylate Thr172 in response to a variety of signals [15,16]. Furthermore, Thr172 can be phosphorylated by CAMKK2 (Calcium/calmodulin-dependent protein kinase kinase 2) kinase in response to calcium flux, independently of LKB1 [17]. Several other studies have suggested that the MAPKKK family member TAK1/MAP3K7 (Transforming growth factor beta-activated kinase 1)/(Mitogen-activated protein kinase 7) might also phosphorylate Thr172 [18,19]. Apart from the direct regulation of AMPK activity by phosphorylation, its γ-subunit acts a sensor that enables AMPK to respond to changes in AMP/ATP or ADP/ATP levels [20]. Moreover, AMP alone is able to directly modulate AMPK activity through three distinct mechanisms. First, AMP may stimulate the phosphorylation of Thr172 by acting directly on upstream kinases [21]. Second, through allosteric modulation, AMP allows AMPK to be a more attractive substrate for its upstream kinases [22]. Third, AMP could inhibit Thr172 dephosphorylation by protecting it from phosphatases activity, or also increase AMPK activity once Thr172 is phosphorylated, also in an allosteric fashion [23,24].

## 3. AMPK as an Energy-Sensing Kinase for Metabolic Regulation

AMPK is one of the main energy-sensing kinases in eukaryotic cells, able to regulate a wide variety of metabolic processes either by directly acting on metabolically-relevant proteins or by indirectly influencing gene expression [4,25]. During energy stress, AMPK directly activates metabolic enzymes and regulates different energy-consuming/producing pathways, (i.e., lipid or glucose metabolism, or mitochondrial biogenesis) in order to maintain an adequate energy balance. Moreover, AMPK can inhibit the activity of several transcription factors involved in anabolic routes, such as lipid, protein, and carbohydrate biosynthesis, to minimize ATP consumption. By contrast, AMPK promotes the activity of numerous transcription factors implicated in catabolism pathways to stimulate ATP production such as glucose uptake and metabolism (Figure 2) [26].

### 3.1. Lipid Metabolism

AMPK is a major regulator of cellular lipid metabolism. Once activated, AMPK is able to reduce the activity of essential enzymes for lipid synthesis and related processes, in order to couple their activity to cell energy levels. One of the best-studied AMPK functions in this context is the inhibitory phosphorylation of ACC1 (acetyl-CoA carboxylase 1) and ACC2 (acetyl-CoA carboxylase 2), which catalyze the first step in lipid synthesis. Several studies using animal models have shown that AMPK-mediated phosphorylation of Ser79 in ACC1 and Ser221 in ACC2 are mechanistically involved in the regulation of lipid homeostasis by AMPK [27]. Similarly, AMPK inhibits HMGCR (3-hydroxy-3-methylglutaryl-coenzyme A reductase), which catalyzes an essential step in cholesterol synthesis [28]. Conversely, AMPK positively promotes triglyceride conversion to fatty acids by stimulating lipases such as ATGL (adipocyte triglyceride lipase) and HSL (hormone-sensitive lipase) [29]. When cellular energy is low, free fatty acids can be imported into mitochondria for β-oxidation. This process requires the activity of the acyl-transferases form the CPT1 (Carnitine palmitoyl-transferase 1) family [30]. AMPK indirectly participates in the regulation of CPT1 because malonyl-CoA generated by ACC1 and ACC2 is a potent inhibitor of CPT1 activity. Therefore, AMPK activity directly decreases lipid synthesis and indirectly increases fatty acid import into mitochondria for β-oxidation [31,32].

In addition to its direct activity, AMPK is able to modulate lipid metabolism by regulating the activity of several transcription factors involved in lipid synthesis and associated processes. Specifically, AMPK phosphorylation inhibits the transcriptional activity of SREBP1 (sterol regulatory element binding protein 1), ChREBP (carbohydrate-responsive element binding protein) or HNF4α (hepatocyte nuclear factor 4α) [33,34,35,36]. Thus, by inhibiting phosphorylation of these transcription factors, AMPK negatively regulates lipogenic processes through the modulation of lipid-specific transcriptional programs.

### 3.2. Glucose Metabolism

In parallel to its inhibitory role for lipid synthesis, AMPK activity contributes to ATP generation through the modulation of a variety catabolic and anabolic pathways involved in glucose metabolism [37]. AMPK promotes glucose uptake by its inhibitory phosphorylation on TBC1D1 (TBC domain family member 1) and TXNIP (thioredoxin-interacting protein). These two factors respectively inhibit translocation of glucose transporters GLUT1 (Glucose transporter 1) and GLUT4 (Glucose transporter 4) to the plasma membrane. Thus, a high AMPK activity is associated to an increased presence of GLUT1 and GLUT4 glucose transporters in the plasma membrane [38,39]. Consistently, AMPK positively regulates glycolysis by phosphorylating PFKFB3 (6-phosphofructo-2-kinase/fructose-2,6-biphosphatase) [40]. Moreover, AMPK also inhibits glucose conversion into glycogen by inhibitory phosphorylation on different isoforms of GYS (glycogen synthase) [41,42]. Paradoxically, AMPK is involved in the regulation of glycogen supercompensation in skeletal muscle. Under conditions of prolonged physical exercise, sustained activation of AMPK boosts glycogen synthesis, especially in skeletal muscle. This is produced as a result of an increase in glucose uptake, which leads to an accumulation of intracellular G6P (glucose 6 phosphate) that allosterically activates the GYS, thus bypassing the inhibitory action of AMPK on this enzyme [43,44]. In parallel to its direct action on specific enzymes, AMPK also modulates glucose metabolism in a transcriptional fashion. This is the case for gluconeogenesis, which is activated during fasting or reduced glucose intake to maintain blood glucose levels. After re-feeding, a surge in insulin levels leads to phosphorylation of liver AMPK by either AKT (RAC-alpha serine/threonine-protein kinase) or LKB1 kinases, which in turn leads to transcriptional inhibition of gluconeogenesis key genes. This effect is partially achieved by AMPK-dependent phosphorylation and nuclear exclusion of CRTC2 (cyclic-AMP-regulated transcriptional co-activator 2) and HDACs (class IIA histone deacetylases), which are all essential co-factors for the transcription of gluconeogenic genes [45,46]. Thus, AMPK can influence glucose metabolism either by direct regulation of specific proteins or by transcriptional regulation of key genes involved in glucose metabolism [47].

### 3.3. Mitochondrial Biogenesis

AMPK activity is also important for maintaining mitochondrial function. In situations of energy imbalance, AMPK contributes to mitochondrial biogenesis in order to increase ATP production. During this process, cells increase their individual mitochondrial mass, which requires an increase in the expression of mitochondrial protein genes [48]. One of the main regulators of this process is PGC1-α (peroxisome proliferator-activated receptor-gamma coactivator), which transcriptionally controls the expression of a wide variety of mitochondrial genes. Overexpression of PGC1-α in muscle contributes to the conversion of type IIb fibers into type II and type I fibers, which are rich in mitochondria [49]. PGC1-α interacts with PPAR-γ (peroxisome proliferator-activated receptor-γ) and ERRs (estrogen-related receptors) and is regulated by numerous post-translational mechanisms [50]. These mechanisms include methylation, acetylation, and phosphorylation by upstream kinases. Different in vitro studies indicate that PGC1-α harbors two sites susceptible of being phosphorylated by AMPK, specifically Thr177 and Ser538. In addition, AMPK can indirectly regulate PGC1-α through phosphorylation of additional targets, such as HDAC5, SIRT1 and p38 MAPK. Moreover, AMPK promotes the activity of TFEB (transcription factor EB), which activates the gene encoding PGC1-α, *PPARGC1A,* as well as different genes involved in autophagy [51,52,53,54].

AMPK-dependent mitochondrial biogenesis has been specifically studied in skeletal muscle in response to exercise. Exercise activates AMPK in myocytes, leading to mitochondrial biogenesis upregulation [55]. Several studies have shown that overexpression of a constitutively active AMPK γ3-subunit induces mitochondrial biogenesis in mice [56]. In addition, AMPK activation improves muscle regeneration and protects muscle from age-related pathologies, in part by increasing autophagic activity [57].

## 4. AMPK: Regulation of Autophagy

### 4.1. Autophagy Regulation

Autophagy is an essential catabolic pathway conserved in all known nucleated cells [7,58]. Although autophagic degradation is constitutively active at basal levels, the main physiological autophagy inducer is nutrient deprivation and/or energy scarcity. This intrinsic characteristic has remained unaltered in organisms ranging from yeast to humans, which makes the involvement of AMPK in the regulation of this process logical.

An autophagy pathway starts with the formation of double-membrane vesicles called autophagosomes. These autophagosomes sequester cytoplasmic cargo (both through specific and non-specific mechanisms) and move along the cellular microtubule network until they eventually fuse with lysosomes [59]. Autophagosome-lysosome fusion allows for the degradation of autophagosome cargo and the autophagosomal inner membrane. Once degradation has occurred, the resulting biomolecules, such as amino acids, lipids, or nucleotides are recycled back to the cytoplasm and will be reused by the cell to synthesize new biomolecules [60]. From a molecular perspective, the autophagy pathway requires the involvement of a group of evolutionarily conserved genes/proteins called ATG (AuTophaGy-related) proteins [61]. These proteins were originally described in yeast and are required for autophagosome formation, maturation, transport, or degradation, being involved in the different steps of the autophagic pathway in a hierarchical and temporally-coordinated fashion (Figure 3) [61,62].

In mammalian cells, autophagosome biogenesis requires the combined activity of two protein complexes, namely the Class III PI3-Kinase protein complex and ULK1/2 (unc-51-like kinase1/2)-containing complexes, which are recruited to autophagosome forming sites during autophagy initiation [63,64,65]. Mammalian ULK proteins are a family of serine/threonine kinases which are the orthologues of yeast ATG1 and whose activity is essential for the recruitment of autophagy-relevant proteins involved in autophagosome biogenesis [66,67]. There are four members of the ULK family in mammalian cells (ULK1-4). ULK1 is the main ATG1 functional orthologue in mammalian cells [68], although ULK2 has been shown to compensate for ULK1 loss. By contrast, ULK3 and ULK4 seem to have evolved to perform biological functions unrelated to autophagy [68]. ULK1 is part of a protein complex containing ATG13, FIP200, also known as RB1CC1 (RB1-inducible coiled-coil protein 1), and ATG101. As we will discuss later, the autophagy-promoting activity of this protein complex can be modulated through specific phosphorylation of its subunits.

The activity of ULK1 is not enough to promote efficient autophagosome biogenesis. In fact, the activity of the Class III PI3-Kinase protein complex, which contains the catalytic subunit VPS34 (Vacuolar protein sorting 34) and other variable regulatory subunits, is also required for autophagosome formation. This protein complex, which acts as a lipid kinase, generates PIP3P-enriched membrane domains at the site of autophagosome formation, which are required to recruit essential factors for autophagosome formation. Apart from the autophagy-relevant Class III PI3K complex, which is formed by VPS34, VPS15, Beclin1 (Coiled-coil, myosin-like BCL2 interacting protein), ATG14, and AMBRA1 (Activating molecule in Beclin1-regulated autophagy protein 1), VPS34 can form part of other Class III PI3K complexes. These alternative VPS34 protein complexes are comprised of different subunits and regulate vesicular trafficking in processes such as endocytosis or Golgi-mediated protein secretion [69,70]. The relative abundance of these different VPS43-containing complexes is variable and will be regulated according to cellular needs. Thus, in conditions of autophagy induction, most VPS34-containing complexes will consist of autophagy-relevant subunits. Thus, autophagy regulation at this level is achieved both by direct modulation of VPS34 kinase activity and by increasing the formation of autophagy-relevant VPS34-containing complexes.

In addition to the activity of these protein complexes and other ATG proteins, which are specifically involved in autophagy execution, multiple signaling cascades are able to regulate autophagic activity. The relative importance of the different regulatory inputs for autophagy execution is, in many cases, cell type-specific and tissue-dependent. However, there are some major autophagy regulatory circuits which have remained conserved through evolution and are present in most tissues/cell types from most multicellular organisms, including mammals, such as AMPK or mTORC1 [59,71].

### 4.2. AMPK Antagonizes mTORC1 to Regulate ULK Complex Activity

In mammalian cells, although many different pathways or signaling events may influence autophagy, the two main regulators for autophagic degradation are mTOR (mechanistic/mammalian Target Of Rapamycin) and AMPK kinases [72,73]. mTOR can be found forming two protein complexes, mTORC1 and mTORC2. Although mTORC2 involvement in autophagy dynamics is not totally neglectable, its contribution to autophagy regulation is marginal. By contrast, mTORC1 can be considered as the main autophagy suppressor in mammalian cells. mTORC1 is normally active in situations of high energy levels, high amino acid cellular content, or growth factors stimulation, all of which have a negative impact in autophagic degradation [74]. By contrast, AMPK activation positively regulates autophagic activity, as one may expect due to its pro-catabolic functions. Due to their antagonistic roles, mTORC1 and AMPK activities are molecularly connected, inhibition of mTORC1 activity being one of the main mechanisms by which AMPK increases autophagic degradation and vice versa.

When energy/growth factors or amino acids are abundant, mTORC1 represses autophagy through inhibitory phosphorylation of ATG13, which reduces the activity of the ULK1 complex, thus decreasing the rate of autophagosome formation [75,76]. In the same sense, ULK1 itself is a direct target of mTORC1, so mTORC1 can inhibit the autophagic process by acting both on ULK1 and ATG13 [75]. AMPK plays an opposite role to mTORC1 regarding ULK1 complex activity, thus positively regulating the first steps of autophagosome formation in response to a variety of pro-autophagic stimuli [77,78]. In fact, AMPK increases ULK1 activity by directly phosphorylating Ser467, Ser555, Thr574, and Ser637, which increases the recruitment of autophagy-relevant proteins (ATG proteins) to the membrane domains in which autophagosome formation takes place [79].

In addition, AMPK negatively regulates mTORC1 activity, which blocks its inhibitory effect on ULK1 by two complementary actions [74]. First, AMPK activates TSC2 (Tuberous sclerosis complex 2) by phosphorylating Thr1227 and Ser1345 residues, thus favoring the assembly of TSC1/TSC2 heterodimer, which negatively impacts mTORC1 activity [80]. Second, AMPK can inhibit mTORC1 by direct phosphorylation of RAPTOR (regulatory-associated protein of mTOR) Ser722 and Ser792 residues [81]. In addition, AMPK is able to promote autophagy by acting differentially at different levels of autophagy regulation, through specific phosphorylation in components of autophagy-initiating protein complexes. Again, the activities of AMPK and mTORC1 are antagonistic in relation to autophagy regulation, and they act together to couple autophagy regulation with multiple signaling pathways, with the ULK1 complex being one of the main checkpoints for the regulation of autophagy initiation (Table 1).

### 4.3. AMPK Regulates Class III PI3K Complex Activity

Apart from its role in regulating ULK1 activity, AMPK is also involved in the regulation of Class III PI3K complex activity. ULK1 itself exerts its pro-autophagic activity by phosphorylating several components of this complex, including Beclin1, AMBRA1, or the catalytic subunit VPS34 [65]. Interestingly, mTORC1 also inhibits autophagosome biogenesis through phosphorylation of ATG14L, an essential component of the pro-autophagic VPS34 complex [64].

AMPK regulation of autophagy also operates through phosphorylation in different subunits of the different Class III PI3K complexes, including VPS34 itself, which modifies their affinity for other components of the complex. Thus, AMPK regulates the relative abundance of the different VPS34-containing complexes, thus connecting the activity of processes involving vesicle trafficking to cellular energy status. In this regard, different biochemical studies have shown how AMPK regulates the composition of the Class III PI3K complex. For example, AMPK phosphorylation of Beclin1 at Thr388 increases Beclin1 binding to VPS34 and ATG14, which promotes higher autophagy activity upon glucose withdrawal than the wild-type control [83,88]. Similarly, AMPK phosphorylation of mouse Beclin1 at Ser-91 and Ser-94 increases the rate of autophagosome formation under nutrient stress conditions [83]. Apart from its activity towards components of the different Class III PI3K complexes, AMPK can also influence their composition by phosphorylating other proteins, which are relevant for the formation/stability of VPS34-containing complexes. Thus, AMPK-mediated phosphorylation of Thr32 on PAQR3 (progestin and adipo-Q receptors member 3), an ATG14L/VPS34 scaffolding protein, or that of Thr50 on the VPS34 associated protein RACK1 (Receptor for activated C kinase 1) has also been shown to enhance stability and pro-autophagic activity of Class III PI3K complexes [85,86].

In parallel to its activating phosphorylation in diverse components of the pro-autophagic VPS34 complexes, AMPK inhibits VPS34 complexes that do not contain pro-autophagic factors and are thus involved in different cellular vesicle trafficking processes, by direct phosphorylation of VPS34 on Thr163 and Ser165 [83]. Hence, in autophagy-promoting conditions, AMPK activation both enhances the activity of pro-autophagic VPS34 complexes and inhibits the formation of other different Class III PI3K complexes involved in autophagy-independent processes.

### 4.4. Additional AMPK Regulation of Autophagy

Apart from its direct activity towards autophagy-initiating complexes, AMPK can also influence autophagic activity by specific phosphorylation of ATG9, a transmembrane protein involved in autophagosome biogenesis by supplying vesicles which contribute to autophagosome elongation. In fact, AMPK-mediated phosphorylation in Ser761 of ATG9 increases recruitment of ATG9A (and ATG9-containing vesicles) to LC3-positive autophagosomes, thus enhancing autophagosome biogenesis [82].

Additionally, AMPK is able to influence autophagic activity in a transcriptional fashion (Table 2). In fact, under stress situations, AMPK directly phosphorylates the FOXO3 (Forkhead box O3) transcription factor, which regulates genes implicated in autophagy execution [89]. This activity antagonizes that of the mTOR, which on the other hand phosphorylates other members of the FOX (Forkhead box) family, such as FOXK2 (Forkhead box protein K2) and FOXK1 (Forkhead box protein K1), which compete with FOXO3 to repress genes implicated in autophagy [90]. A similar situation in which AMPK and mTOR activities antagonize each other can be found in relation with TFEB/TFE transcription factors, which control the expression of a variety of genes involved in lysosomal biogenesis and autophagy [91].

In situations of high energy, mTOR phosphorylates these factors of transcription and inhibits their function. By contrast, recent reports have shown that TFEB/TFE nuclear translocation is highly reduced either in cells deficient for AMPK or treated with AMPK inhibitors [92,93]. Consistently, it has been recently reported that AMPK activity is required for efficient dissociation of the transcriptional repressor BRD4 (Bromodomain-containing protein 4) from autophagy gene promoters in response to starvation [94].

Apart from its ability to regulate autophagy-relevant transcription factors through direct phosphorylation, AMPK also regulates different transcriptional regulators, such as EP300 [95] or Class IIa HDACs [47], which are involved in metabolism, autophagy and lysosomal functions (Table 2).

### 4.5. Selective Degradation of Mitochondria by Autophagy

Apart from its general role in the regulation of bulk autophagic degradation, AMPK specifically participates in the regulation of mitophagy, the selective elimination of defective mitochondria through autophagy [101]. Mitochondria are organized in a dynamic network that changes its morphology through the combined actions of fission and fusion. Thus, mitochondria can be found in different distributions ranging from a single closed network to large numbers of small fragments. Recent studies have shown that an increase in mitochondrial fission is required in order to facilitate mitophagy [102]. This renders mitochondria susceptible to being engulfed by pre-autophagosomal isolation membranes, thus allowing mitophagy to take place. Consistently, mitochondrial stressors, such as electron transport chain poisons, or other stressors that damage mitochondria (and thus would increase mitophagy) have been shown to increase mitochondrial fission [103]. AMPK activation by energy imbalance and also a variety of other cellular and mitochondrial stressors promotes mitochondrial fission, thus coupling mitochondrial dynamics with mitophagic degradation [104]. This effect mainly relies on the ability of AMPK to phosphorylate and to activate the MFF (mitochondrial fission factor). MFF is a mitochondrial outer-membrane protein, which recruits cytoplasmic DRP1 (dynamin 1 like protein) to the mitochondrial outer membrane [105]. DRP recruitment to the mitochondrial outer-membrane increases mitochondrial fission, enabling the resulting fragmented mitochondria to undergo mitophagy [106].

In this context, AMPK phosphorylation of ULK1 on Ser555 has been shown to be critical for the development of exercise-induced mitophagy [107]. Thus, through its combined actions on key factors for autophagy regulation (by acting on ULK1 and other major autophagy regulators) and mitochondrial network dynamics (by specifically activating MFF), AMPK is able to coordinate autophagosome formation and mitochondrial size in order to enable efficient autophagic degradation of mitochondria.

## 5. Conclusions and Future Perspectives

Autophagy regulation has become increasingly complex in high eukaryotes, in which multiple signaling cascades are connected to autophagy key factors. However, AMPK probably remains as the major molecular autophagy inducer, counteracting the activity of mTORC1, which has also evolutionarily remained as the main molecular autophagy inhibitor. In parallel to its function in autophagy regulation, AMPK’s role of adapting cellular metabolism to energetic availability has also been conserved in a diversity of organisms, from yeast to mammals. Thus, it is not surprising that substantial efforts have been made to identify new pharmacological AMPK activators. Recently, a variety of compounds able to increase AMPK activity, such as AICAR, Compound-13, PT-1, A769662 or benzimidazole have been identified [108]. Many of these drugs have shown great potential as research tools to modulate AMPK activity, and some of them have been successfully tested in animal models. However, and despite the substantial advances in this field, metformin (clinically developed in the late 1950s) is still the only AMPK-activating drug widely used in human patients. Interestingly, therapeutic AMPK modulation by metformin has shown promising results in the context of diverse metabolic conditions such as type II diabetes, fatty liver diseases, Alzheimer’s, and in diverse types of cancers [109]. In fact, the beneficial effects of metformin are sometimes beyond the scope of its a priori potential. The fact that AMPK plays a pivotal role in autophagy regulation, together with the wide variety of processes for which autophagic activity is beneficial, suggests a potential mechanistic involvement of autophagy for some of the positive effects of AMPK activation. Future studies aimed at dissecting the precise molecular mechanisms by which AMPK exerts its wide variety of beneficial effects for human health will shed more light into these questions.

## Figures and Tables

**Figure 1 ijms-19-03812-f001:**
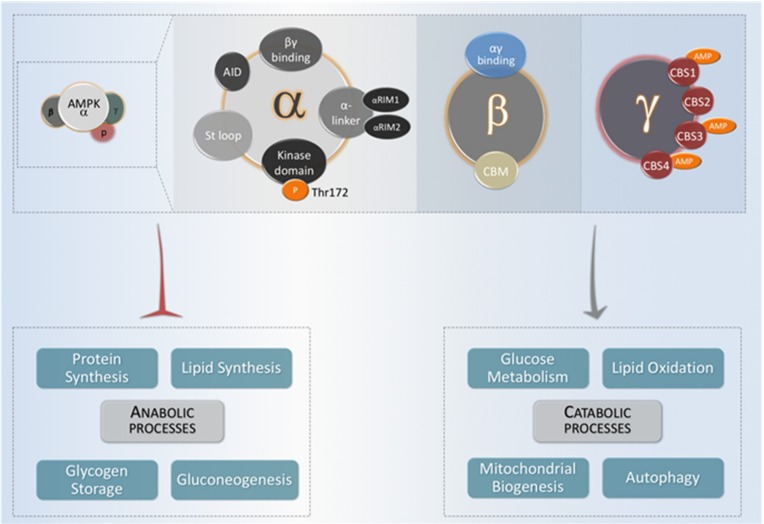
The domain structure of AMPK (adenosine monophosphate-activated protein kinase) heterotrimer. Functional AMPK complexes consist of one catalytic and two regulatory subunits. When activated, AMPK acts by decreasing energy-consuming anabolic processes (lipid synthesis, glycogen storage, gluconeogenesis, and protein synthesis) and increasing energy-providing catabolic processes that provide ATP (glucose metabolism, lipid oxidation, mitochondrial biogenesis and autophagy).

**Figure 2 ijms-19-03812-f002:**
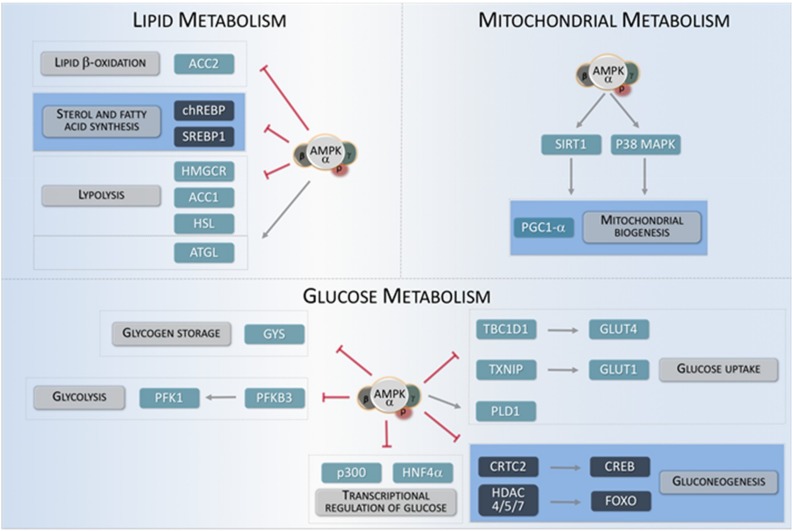
AMPK regulates different metabolic key targets. The metabolic pathways modulated by AMPK can be classified into three general categories: Lipid metabolism, mitochondrial metabolism, and glucose metabolism. The arrow indicates key targets for AMPK involved in these three metabolic categories. Transcriptional regulators are shown in dark squares.

**Figure 3 ijms-19-03812-f003:**
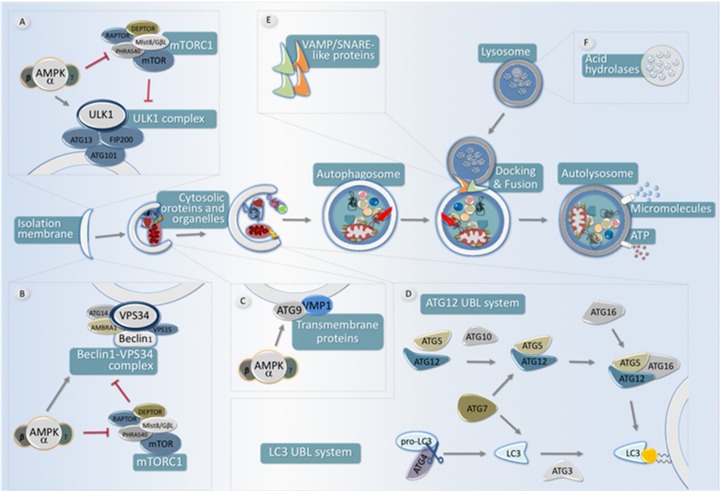
Scheme of the main steps for autophagy and their regulation by AMPK. An autophagy pathway starts with the formation of the isolation membrane, also known as phagophore. The autophagy implicates the coordinated temporal and spatial activation of numerous molecular components. (**A**): The ULK1-FIP200-ATG13-ATG101 complex is responsible for initiating the autophagic process. The activity of this protein complex is antagonistically regulated by mTORC1 (inhibitory phosphorylation) and by AMPK, which both activates the ULK1 complex as well as inhibits the activity of the mTORC1 complex. (**B**): The ClassIII PI3K complex formed by VPS34, Beclin1, ATG14, AMBRA1, and other subunits creates a membrane domain enriched in PtsIns3P, which drives the nucleation of ATG (AuTophaGy-related) proteins in the phagophore, either directly or indirectly. AMPK is able to increase the pro-autophagic function of this complex and to enhance its formation, whereas mTORC1 activity negatively regulates its function. (**C**): Two different transmembrane proteins, the vacuole membrane protein 1 (VMP1) and ATG9 participate in the recruitment of membranes to the phagophore. AMPK is able to phosphorylate ATG9, which increases its recruitment towards autophagosome formation sites. (**D**): Two ubiquitin-like (UBL) protein conjugation systems (ATG12- and LC3- UBLs) involving the participation of ATG4 cysteine proteinases (which activate LC3 by cleaving its carboxyl terminus), the E1-like enzyme ATG7 (common to both conjugation systems), and the E2-like enzymes ATG10 (ATG12 system), and ATG3 (LC3 system). In coordination, the activity of both systems is required to conjugate LC3 (and other members of this protein family homologous to yeast ATG8) to a phosphatidyl-ethanolamine lipid at the nascent pre-autophagosomal membrane. (**E**): Upon completion, fully-formed autophagosomes move along the microtubule network, eventually fusing with a lysosome, thus acquiring hydrolytic activity, and thus becoming autolysosomes. Several SNARE-like proteins (i.e., Syntaxin17 and VAMP8, among others) are required for efficient fusion between lysosomes and autophagosomes. Once content and inner membrane are degraded by acidic hydrolases, the resultant molecules (amino acids, nucleotides, lipids, etc.) are recycled back to the cytoplasm by membrane permeases.

**Table 1 ijms-19-03812-t001:** Regulation of autophagy relevant proteins by AMPK. H, human; M, mouse; R, rat.

Protein	Phosphorylation Site(s)	Stage of Autophagy	Autophagy Function	Ref.
ATG9	Ser761(H, M, R)	Autophagosome elongation	Participates in the recruitment of lipids to the isolation membrane	[82]
BECN1	Ser91(M, R) Ser94(M, R)	Autophagosome biogenesis	Part of the III PI3KC3 complex	[83]
mTOR (RAPTOR)	Ser722(H, M) Ser792(H, M)	Regulation of Autophagy	Negative regulator of Autophagy	[81]
mTOR	Thr2446(H)	Regulation of Autophagy	Negative regulator of Autophagy	[84]
PAQR3	Thr32(H, M)	Autophagosome biogenesis	Facilitates the formation of pro-autophagic PI3KC3 III complex	[85]
RACK1	Thr50(H, M, R)	Autophagosome biogenesis	Promoting the assembly of the III PI3KC3 complex	[86]
TSC2	Ser1387(H, M, R) Thr1271(H, R)	Regulation of Autophagy	Negative regulator of Mtor	[80,87]
ULK1	Ser555(M, R) Ser467(H, M, R) Thr574(M, R) Ser637(M, R)	Autophagy Initiation	Part of the ULK1-complex/early steps of autophagosome biogenesis	[79]
VPS34	Thr163(H, M, R) Ser165(H, M, R)	Autophagosome biogenesis	Part of the III PI3KC3 complex	[83]

**Table 2 ijms-19-03812-t002:** Transcriptional regulation of autophagy through AMPK phosphorylation. H, human; M, mouse; R, rat.

Transcription Factor	Phosphorylation Site(s)	Target Gene (s)	Ref.
CHOP	Ser30(H, M, R)	*ATG5, MAP1LC3B*	[96]
FOXO3	Thr179(H) Ser399(H) Ser413(H) Ser555(H) Ser588(H) Ser626(H)	*ATG4B, GABARAPL1, ATG12, ATG14, GLUL, MAP1LC3,* *BECN1, PIK3CA, PIK3C3, ULK1, BNIP3, FBXO32*	[89]
HSF1	Ser121(H, M, R)	*ATG7*	[97]
Nrf2	Ser558(H, M)	*SQSTM1*	[98]
p53	Ser15(H, R)	*AEN, DRAM1, BAX, IGFBP3, BBC3, C12orf5, PRKAB1, PRKAB2, CDKN2A, SESN1, SESN2, DAPK1, BCL2, MCL1*	[99]
p73	Ser426(H)	*ATG5, DRAM1, ATG7, UVRAG*	[100]

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
