# Peer review of "AMPK: Regulation of Metabolic Dynamics in the Context of Autophagy"

_ijms, 2018, doi:10.3390/ijms19123812_

Round 1
Reviewer 1 Report
A very informative and indepth review of AMPK and autophagy. The review will benefit scientists focussed on this exciting area of research and provides them with a concise and thorough review of the current understanding within the field.
Revisions:
Please review Figure 1. Firstly, the figure is misleading in that the beta and gamma subunit look like they are fused (joined together). This is misleading to readers since they are separate proteins. Furthermore, for me this is too similar (almost identical) to Figure 1 in a recent review article by Herzig et al. 2018, published in Nat Rev Mol cel Biol. 19(2): 121-135. doi:10.1038/nrm.2017.95, from Reuben Shaw`s lab. I noticed this because its such a distinctive way of depicting the subunits together, which I think is the correct way of displaying the subunits.
Please check the spelling of Glycolysis and Lipolysis in your figures and text. In a number of places the authors spell it Glycolisis and Lipolisis.
Author Response
We acknowledge the reviewer for her/his constructive comments that were useful to improve our first version of this manuscript. We have addressed all the reviewer’s points, as discussed below:
Please review Figure 1. Firstly, the figure is misleading in that the beta and gamma subunit look like they are fused (joined together). This is misleading to readers since they are separate proteins. Furthermore, for me this is too similar (almost identical) to Figure 1 in a recent review article by Herzig et al. 2018, published in Nat Rev Mol cel Biol. 19(2): 121-135. doi:10.1038/nrm.2017.95, from Reuben Shaw`s lab. I noticed this because its such a distinctive way of depicting the subunits together, which I think is the correct way of displaying the subunits.
Response:We have complied with this request, we have modified figure 1 and the style is completely different from the figure that appears in the review article by Herzig et al. 2018, published in Nat Rev Mol cel Biol. 19(2): 121-135. doi:10.1038/nrm.2017.95, from Reuben Shaw`s lab. We thank the reviewer for his comment.
Please check the spelling of Glycolysis and Lipolysis in your figures and text. In a number of places the authors spell it Glycolisis and Lipolisis.
Response :According to this reviewer, we have replaced Glycolisis and Lipolisis with Glycolysis and Lipolysis in all places. We thank the reviewer for his comment.
Reviewer 2 Report
There are two major issues with this review article:
(1) Under lines 299-301 and Table 1, the listed ULK1 phosphorylation sites (S555, S467, T574, S637) did not match to the listed two references (76, 82). The Wang et al paper (82) is a review paper. The Kim et al paper (76) actually describes different phosphorylation sites under glucose starvation and nutrient sufficiency conditions. That is, under starvation, AMPK directly activates ULK1 by S317/S777 phosphorylation while under nutrient sufficient, mTOR phosphorylates ULK1 S757 which results in disruption of AMPK-ULK1 interaction. The four phosphorylation sites mentioned in the manuscript and listed in Table 1 were discovered in a paper by Egan et al (Science. 2011 Jan 28; 331(6016): 456–461. Phosphorylation of ULK1 (hATG1) by AMP-activated protein kinase connects energy sensing to mitophagy).
(2) Although mitochondrial biogenesis was reviewed (section 3.3), the selective degradation of mitochondria by autophagy also needs to be discussed as well. AMPK-induced mitochondria fragmentation has become an increasingly topic and a better discussion would benefit researcher in the field. The following is a good starting reference for such discussion: Toyama EQ et.al Science. 2016 Jan 15; 351(6270): 275–281. Metabolism. AMP-activated protein kinase mediates mitochondrial fission in response to energy stress.
Author Response
We acknowledge the reviewer for her/his constructive comments that were useful to improve our first version of this manuscript. We have addressed all the reviewer’s points, as discussed below:
There are two major issues with this review article:
(1) Under lines 299-301 and Table 1, the listed ULK1 phosphorylation sites (S555, S467, T574, S637) did not match to the listed two references (76, 82). The Wang et al paper (82) is a review paper. The Kim et al paper (76) actually describes different phosphorylation sites under glucose starvation and nutrient sufficiency conditions. That is, under starvation, AMPK directly activates ULK1 by S317/S777 phosphorylation while under nutrient sufficient, mTOR phosphorylates ULK1 S757 which results in disruption of AMPK-ULK1 interaction. The four phosphorylation sites mentioned in the manuscript and listed in Table 1 were discovered in a paper by Egan et al (Science. 2011 Jan 28; 331(6016): 456–461. Phosphorylation of ULK1 (hATG1) by AMP-activated protein kinase connects energy sensing to mitophagy).
Response: According to the reviewer’s comment, we have replaced the two cited references (Wang et al and Kim et al articles) for Egan et al under lines 299-301 and Table 1. We thank the reviewer for his comment, this reference is more accurate than the references we had previously cited.
(2) Although mitochondrial biogenesis was reviewed (section 3.3), the selective degradation of mitochondria by autophagy also needs to be discussed as well. AMPK-induced mitochondria fragmentation has become an increasingly topic and a better discussion would benefit researcher in the field. The following is a good starting reference for such discussion: Toyama EQ et.al (Science. 2016 Jan 15; 351(6270): 275–281. Metabolism. AMP-activated protein kinase mediates mitochondrial fission in response to energy stress)
Response: According to this reviewer, we have incorporated the section 4.5 "selective degradation of mitochondria by autophagy" in which we deal with the topic of mitophagy and the role played by AMPK. We have incorporated the reference suggested by the reviewer in our references, in addition to some other key references of recent publication. We thank the reviewer for his comment, we believe that this change benefits the scientific content of our article.
Reviewer 3 Report
This review by Tamargo-Gómez and Mariño discusses the metabolic dynamics regulated by AMPK and the molecular links between AMPK signaling and autophagy regulation with a focus on the phosphorylation of autophagy-related proteins. Overall this review is well-written with several figures and tables that help guide readers through the literature being reviewed. To improve the manuscript’s overall focus and clarity, major and minor comments to the authors are provided below that include several suggested figure and table modifications as well as minor text revisions.
Major Comments:
1. In Figure 1 it is not clear where the AMPK beta subunit ends and the gamma subunit begins. Consider using different colors for each of these subunits or more clearly separate these subunits from each other to improve figure clarity. In addition, in Figure 2 the authors should consider clearly labeling the nuclear compartment or nuclear-related proteins if they want to more clearly highlight nuclear receptors and transcription factors within each main box.
2. In Figures 1, 2 and 3, consider increasing the size of text boxes and/or the font size within these text boxes to improve clarity and ensure this text will be easily read and stand out from the text boxes even when the manuscript is printed in black and white. There is almost as much empty space as space being used for diagrams and text, so additional space is available without changing the overall figure size.
3. The list of abbreviations below each figure disrupts the overall flow of the manuscript and distracts the reader from the main text. In line with the recommended formatting for IJMS manuscripts, the compiled abbreviations should appear together at the end of the manuscript before the reference list.
4. The sentence in Lines 158-161 is not completely clear. The authors should specify what they mean by the phrase ‘glucose generation and storage.’ Moreover, to ensure accuracy and clarity in Lines 166-167 the authors should consider briefly describing the paradoxical relationship between AMPK and glycogen synthase activation in skeletal muscle (i.e. although AMPK phosphorylates and inhibits glycogen synthase in skeletal muscle, repeated AMPK activation such as with exercise training increases skeletal muscle glycogen synthesis due to AMPK’s stimulatory effect on glucose uptake leading to intracellular G6P accumulation that allosterically activates glycogen synthase and overrides AMPK inhibitory role on glycogen synthase).
5. A major concern from this reviewer is that although this review is meant to be focused on AMPK with respect to autophagy, the broad introductory section on AMPK regulation of metabolism is very lengthy (Pages 1-6), and details regarding autophagy regulation by AMPK are not discussed until at least halfway through the review on Page 6. Although Section 4 focusing on autophagy is well-written, the authors should consider trimming down their introduction so they can focus on their main topic of AMPK and autophagy earlier in the manuscript.
6. Given that phosphorylation sites are not always conserved between species (e.g. from mouse and rat to human), to ensure the correct interpretation of tables the species for each phosphorylation site position in Tables 1 and 2 should be listed. Alternatively, if one reference species is referred to in an entire figure, this information can be listed in the figure legend.
7. The sentence in lines 379-381 should be revised to either introduce the development of more recent AMPK activators that have been tested in animal models including non-human primates, or to highlight that the pharmacological activators of AMPK other than metformin that are currently listed (AICAR, A769662) are primarily used as research tools to modulate AMPK activity and are not widely used to treat human patients. As is currently written, this sentence could be misleading to readers not familiar with pharmacological AMPK activators other than metformin and the potential risks associated with their use in human patients.
Minor Comments:
Line 52: Change the word autophagy to autophagy’s.
Line 54: Consider using etc. instead of ‘…’ or specifically list the other building blocks that the authors are referring to in this sentence.
Line 67: Change the word variants to isoforms.
Lines 69-75: To simplify this section and improve clarity for the readers, define the abbreviation AMPK at its first use and then use the abbreviated form throughout the remainder of the manuscript.
Line 104: In the Figure 1 legend, change the word trimer to heterotrimer, as this terminology is most commonly used when referring to AMPK.
Line 114: Remove the word ‘of’ to improve clarity of this sentence.
Line 121: Consider replacing the word routes with pathways to ensure understanding by the readership.
Lines 170 and 179: Consider replacing the word keep/keeping with maintain/maintaining.
Line 205: Change the word form to from.
Lines 207 and 213: Consider revising the word process to processing to improve clarity.
Line 218: Consider starting the second sentence with the word ‘The’ before ‘autophagy…’
Line 226: Insert the word ‘in’ between ‘proteins’ and ‘the phagophore.’
Line 229: Remove the word ‘that.’
Author Response
We acknowledge the reviewer for her/his constructive comments that were useful to improve our first version of this manuscript. We have addressed all the reviewer’s points, as discussed below:
Major Comments:
1. In Figure 1 it is not clear where the AMPK beta subunit ends and the gamma subunit begins. Consider using different colors for each of these subunits or more clearly separate these subunits from each other to improve figure clarity. In addition, in Figure 2 the authors should consider clearly labeling the nuclear compartment or nuclear-related proteins if they want to more clearly highlight nuclear receptors and transcription factors within each main box.
Response: We have complied with this request, we have modified figure 1 and the style is completely different. We use different colors for each subunits and we separate these subunits. In figure 2, we mark more clearly nuclear receptors and transcription factors. We thank the reviewer for his comment, we believe that these changes benefit the aesthetic part of our article.
2. In Figures 1, 2 and 3, consider increasing the size of text boxes and/or the font size within these text boxes to improve clarity and ensure this text will be easily read and stand out from the text boxes even when the manuscript is printed in black and white. There is almost as much empty space as space being used for diagrams and text, so additional space is available without changing the overall figure size.
Response: We have complied with this request, we have modified the size of text boxes to improve clarity in Figures 1,2 and 3. We thank the reviewer for his comment, we agree with you previously the texts had a size too small and we think that these changes benefit the clarity of the figures in our article.
3. The list of abbreviations below each figure disrupts the overall flow of the manuscript and distracts the reader from the main text. In line with the recommended formatting for IJMS manuscripts, the compiled abbreviations should appear together at the end of the manuscript before the reference list.
Response: We have complied with this request, all abbreviations appear together at the end of the manuscript before the reference list. We thank the reviewer for his comment, we believe that this modification will improve the understanding of our article.
4. The sentence in Lines 158-161 is not completely clear. The authors should specify what they mean by the phrase ‘glucose generation and storage.’ Moreover, to ensure accuracy and clarity in Lines 166-167 the authors should consider briefly describing the paradoxical relationship between AMPK and glycogen synthase activation in skeletal muscle (i.e. although AMPK phosphorylates and inhibits glycogen synthase in skeletal muscle, repeated AMPK activation such as with exercise training increases skeletal muscle glycogen synthesis due to AMPK’s stimulatory effect on glucose uptake leading to intracellular G6P accumulation that allosterically activates glycogen synthase and overrides AMPK inhibitory role on glycogen synthase).
Response: We have complied with this request, we have eliminated the phrase ‘glucose generation and storage’ because it was not completely clear, and the sentence did not substantially contribute to the message of the article. On the other hand, we have incorporated a brief explanation of the paradoxical relationship between glycogen synthase activation and AMPK in skeletal muscle and we use as a reference the article of Janzen et al 2018 published in IJMS. We thank the reviewer for his comment because this incorporation substantially improves the scientific aspect of the article.
5. A major concern from this reviewer is that although this review is meant to be focused on AMPK with respect to autophagy, the broad introductory section on AMPK regulation of metabolism is very lengthy (Pages 1-6), and details regarding autophagy regulation by AMPK are not discussed until at least halfway through the review on Page 6. Although Section 4 focusing on autophagy is well-written, the authors should consider trimming down their introduction so they can focus on their main topic of AMPK and autophagy earlier in the manuscript.
Response: We have complied with this request, we have eliminated 8 lines of the section “2. AMPK: Structure and activation mechanism” to reduce its size and focus in the topic of this article. Furthermore, we have added a section 4.5 "selective degradation of mitochondria by autophagy" to increase the content on autophagy in this article. We thank the reviewer for his comment, we think that this modification allows us to centralize the message of the article.
6. Given that phosphorylation sites are not always conserved between species (e.g. from mouse and rat to human), to ensure the correct interpretation of tables the species for each phosphorylation site position in Tables 1 and 2 should be listed. Alternatively, if one reference species is referred to in an entire figure, this information can be listed in the figure legend.
Response: We have complied with this request, we have incorporated the species for each phosphorylation site position in Tables 1 and 2. We thank the reviewer for his comment, we belive that these changes benefit the clarity of the tables in our article.
7. The sentence in lines 379-381 should be revised to either introduce the development of more recent AMPK activators that have been tested in animal models including non-human primates, or to highlight that the pharmacological activators of AMPK other than metformin that are currently listed (AICAR, A769662) are primarily used as research tools to modulate AMPK activity and are not widely used to treat human patients. As is currently written, this sentence could be misleading to readers not familiar with pharmacological AMPK activators other than metformin and the potential risks associated with their use in human patients.
Response: We have complied with this request, we have reformulated the phrase and clarified the misunderstanding. We thank the reviewer for his comment t, we think that this modification will allow better understand the role of AMPK modulating drugs.
Minor Comments:
Line 52: Change the word autophagy to autophagy’s.
Response: We have complied with this request. We have changed the word autophagy to autophagy’s. We thank the reviewer for his comment.
Line 54: Consider using etc. instead of ‘…’ or specifically list the other building blocks that the authors are referring to in this sentence.
Response: We have complied with this request. We have added etc at the end of the list. We thank the reviewer for his comment.
Line 67: Change the word variants to isoforms.
Response: We have complied with this request. We have changed the word variants to isoforms. We thank the reviewer for his comment.
Lines 69-75: To simplify this section and improve clarity for the readers, define the abbreviation AMPK at its first use and then use the abbreviated form throughout the remainder of the manuscript.
Response: We have complied with this request. We have defined the abbreviation AMPK at its first use. We thank the reviewer for his comment.
Line 104: In the Figure 1 legend, change the word trimer to heterotrimer, as this terminology is most commonly used when referring to AMPK.
Response: We have complied with this request. We have changed the word trimer to heterotrimer. We thank the reviewer for his comment.
Line 114: Remove the word ‘of’ to improve clarity of this sentence.
Response: We have complied with this request. We have removed the word of on line 114. We thank the reviewer for his comment.
Line 121: Consider replacing the word routes with pathways to ensure understanding by the readership.
Response: We have complied with this request. We have changed the word routes to pathways. We thank the reviewer for his comment.
Lines 170 and 179: Consider replacing the word keep/keeping with maintain/maintaining.
Response: We have complied with this request. We have changed the word keep and keeping to maintain and maintaining. We thank the reviewer for his comment.
Line 205: Change the word form to from.
Response: We have complied with this request. We have changed the word form to from. We thank the reviewer for his comment.
Lines 207 and 213: Consider revising the word process to processing to improve clarity.
Response: We have complied with this request. We have changed the word process to pathway because we think that this change is more appropriate. We thank the reviewer for his comment.
Line 218: Consider starting the second sentence with the word ‘The’ before ‘autophagy…’
Response: We have complied with this request. We have started the second sentence on line 218 with the word ‘The’ before ‘autophagy’. We thank the reviewer for his comment.
Line 226: Insert the word ‘in’ between ‘proteins’ and ‘the phagophore.’
Response: We have complied with this request. We have insert the word in between ‘proteins’ and ‘ the phagophore’. We thank the reviewer for his comment.
Line 229: Remove the word ‘that.’
Response: We have complied with this request. We have removed the word that on line 229. We thank the reviewer for his comment.

Round 2
Reviewer 3 Report
The authors have addressed my major and minor comments. The revised versions of the manuscript, figures and tables all read more clearly and have significantly improved this review article.